# Somatic Embryogenesis: A Biotechnological Route in the Production of Recombinant Proteins

**DOI:** 10.3390/biotech14040093

**Published:** 2025-11-26

**Authors:** Marco A. Ramírez-Mosqueda, Jorge David Cadena-Zamudio, Carlos A. Cruz-Cruz, José Luis Aguirre-Noyola, Raúl Barbón, Rafael Gómez-Kosky, Carlos Angulo

**Affiliations:** 1Centro Nacional de Recursos Genéticos-INIFAP, Boulevard de la Biodiversidad No. 400, Rancho las Cruces, Tepatitlán de Morelos 47600, Jalisco, Mexico; marcoa.rm.07@gmail.com (M.A.R.-M.); cadenazamudioj@gmail.com (J.D.C.-Z.);; 2Facultad de Ciencias Químicas, Universidad Veracruzana, Prolongación Oriente 6, No. 1009, Orizaba 94340, Veracruz, Mexico; 3Instituto de Biotecnología de las Plantas, Universidad Central “Marta Abreu” de Las Villas, Carretera a Camajuaní km 5.5, Santa Clara 54830, Villa Clara, Cuba; 4Instituto de Investigaciones de la Caña de Azúcar (INICA Villa Clara), Ranchuelo 10400, Villa Clara, Cuba; 5Immunology & Vaccinology Group and Laboratorio Nacional CONAHCYT de Generación de Vacunas Veterinarias y Servicios de Diagnóstico (LNC-GVD), Centro de Investigaciones Biológicas del Noroeste, S.C., Av. Instituto Politécnico Nacional 195, Playa Palo de Santa Rita Sur, La Paz 23096, Baja California Sur, Mexico

**Keywords:** in vitro culture, genetic transformation, plant regeneration, immunization, vaccination

## Abstract

Somatic embryogenesis (SE) is a morphogenetic pathway widely employed in the commercial micropropagation of plants. This route enables the generation of somatic embryos from somatic tissues, which give rise to complete (bipolar) plants that develop like zygotic embryos. SE can proceed via direct or indirect pathways, and both approaches have been adapted not only for large-scale clonal propagation but also for the regeneration of genetically modified plants. In this context, SE can be harnessed as a versatile platform for recombinant protein production, including vaccine antigens and therapeutic proteins, by combining plant tissue culture with genetic transformation strategies. Successful examples include non-model plants, as *Daucus carota* and *Eleutherococcus senticosus* expressing the cholera and heat-labile enterotoxin B subunits, respectively; *Oryza sativa*, *Nicotiana tabacum*, and *Medicago sativa* producing complex proteins such as human serum albumin (HSA), α_1_-antitrypsin (AAT), and monoclonal antibodies. However, challenges remain in optimizing transformation efficiency, scaling up bioreactor-based suspension cultures, and ensuring proper post-translational modifications under Good Manufacturing Practice (GMP) standards. Recent advances in synthetic biology, modular vector design, and glycoengineering have begun to address these limitations, improving control over transcriptional regulation and protein quality. This review highlights the application of SE as a biotechnological route for recombinant protein production, discusses current challenges, and presents innovative strategies and perspectives for the development of sustainable plant-derived biopharmaceutical systems.

## 1. Introduction

Somatic embryogenesis enables the de novo synthesis of specific cells from a single cell. The plant evolutionary principles underlying this phenomenon have been described in detail [1]. It involves exquisitely coordinated genetic reprogramming, which can be triggered by both internal and external factors and is tightly regulated through epigenetic mechanisms [2]. However, several cellular and molecular mechanisms remain elusive [3]. Technological advances over the past decades have driven significant innovations in somatic embryogenesis for plant propagation. For instance, genome editing has enhanced somatic embryogenesis efficiency, exerting profound effects on the regulatory network pathways that control cell totipotency [4]. Another example is the development of embryogenic cell suspension cultures, which are favored for their advantages, such as rapid cell proliferation and high biomass production [5]. Nonetheless, multiple factors influence successful and efficient somatic embryogenesis, including biological (genotype, cell type), chemical (culture medium composition), and physical (pH, temperature, light) conditions [6].

Since most plants can be regenerated via somatic embryogenesis, this approach has been incorporated into plant genetic transformation protocols. In this context, genetic engineering tools have enabled the production of biopharmaceuticals in plant cells, including recombinant vaccines [7]. For instance, a vaccine prototype against bluetongue disease in sheep was produced in peanut (*Arachis hypogaea* L.) using a somatic embryogenesis-based regeneration system [8]. Similarly, another vaccine prototype based on the *Escherichia coli* heat-labile enterotoxin B subunit was produced in somatic embryos of Siberian ginseng (*Eleutherococcus senticosus*), a medicinal plant [9]. Both examples highlight the relevance of somatic embryogenesis in plant-made vaccine technology. Using this approach, several recombinant vaccines against infectious diseases have been developed to date [10].

However, challenges remain in establishing efficient somatic embryogenesis protocols for vaccine production through recombinant DNA technology. Critical issues such as embryo survival, plantlet regeneration, and transformation efficiency must still be addressed. At the level of in vitro morphogenesis, recent innovations include the optimization of plant growth regulator regimes, the use of morphogenic regulators to increase embryogenic competence, and the application of maturation treatments based on abscisic acid, osmotic agents, and controlled desiccation, all of which improve embryo quality and conversion rates. From the transformation perspective, advances in *Agrobacterium*-mediated systems (refined strains and binary vectors), physical DNA delivery methods, and nanomaterial-based carriers, together with CRISPR-guided targeted integration, are being used to enhance stable transgene insertion and expression. In parallel, SE-derived suspension cultures operated in controlled bioreactors, in combination with rational construct design (codon optimization, synthetic and tissue-specific promoters, strong terminators) and glycoengineered plant lines with human-like glycosylation, are contributing to higher recombinant protein yields and improved product quality. Therefore, in this review, we discuss key aspects of direct and indirect somatic embryogenesis applied to plant-made vaccines, placing them in this emerging technological context. Finally, the main hurdles and opportunities are identified to guide improvements in the current and future development of scalable and affordable plant-based vaccines.

## 2. Somatic Embryogenesis: A Route for Optimization

In vitro morphogenesis refers to the capacity of one or more plant parts to undergo developmental changes in response to phytohormonal and/or environmental stimuli [11]. In plant tissue culture, two primary morphogenetic pathways are recognized: organogenesis and somatic embryogenesis [12,13]. Organogenesis involves the de novo formation of complete organs (typically shoots or roots) from an explant (any plant cell or tissue) [14]. In contrast, somatic embryogenesis entails the formation of a complete plant (shoots with roots) from bipolar structures known as somatic embryos, which develop similarly to zygotic embryos [13]. Both morphogenetic routes are utilized in in vitro propagation, genetic improvement, and the production of secondary metabolites across various plant species. Importantly, the choice of pathway depends on the specific objectives of each study.

### 2.1. Direct Somatic Embryogenesis

Somatic embryogenesis (SE) is an in vitro morphogenetic route characterized by the formation of embryos from somatic tissues [15]. However, tissue responses vary, leading to the classification of SE into direct somatic embryogenesis (DSE) and indirect somatic embryogenesis (ISE) [16]. The DSE is defined by the formation of embryos from isolated or grouped somatic cells without prior callus formation (callogenesis) [17,18] (Figure 1). This approach offers the advantage of producing genetically identical plants (clones), due to their unicellular or multicellular origin. It also facilitates rapid in vitro propagation, mechanization, and automation of clonal propagation through temporary immersion systems [19].

Comparatively, DSE has shown greater efficiency and genetic stability in herbaceous and monocot species where embryogenic competence is naturally high, such as *Phalaenopsis* and *Musa*, whereas woody perennials like *Camellia* and *Paeonia* often exhibit genotype-dependent responses and lower embryogenic frequencies [20,21]. In *Syagrus oleracea*, DSE induction from zygotic embryos promoted the synchronous formation of globular and torpedo stage embryos, with clear vascular isolation and high conversion efficiency to plantlets, thereby improving regeneration uniformity and plantlet survival [15]. Similarly, *Phalaenopsis amabilis* responded strongly to combinations of TDZ and α-naphthaleneacetic acid (NAA); stem explants cultured on this produced an average of 36 embryos per explant under dark preconditioning, demonstrating the high embryogenic potential of this species [22]. In *Camellia oleifera* and banana/plantain (*Musa* spp.), direct and indirect SE pathways have been successfully established from immature seeds, where the combination of 2,4-D and thidiazuron (TDZ) produced bipolar embryos histologically isolated from maternal tissue, confirming the somatic origin rather than organogenic regeneration [23,24]. In contrast, *Paeonia arborea* often exhibits limited embryogenic induction due to phenolic oxidation and explant browning, a common limitation in woody taxa [21]. These comparative findings demonstrate that DSE protocols must be species-specific and highly sensitive to hormonal balance, explant physiology, and culture environment. Nevertheless, when optimized, DSE provides an efficient route for maintaining genetic fidelity and minimizing somaclonal variation, an essential prerequisite for recombinant protein production systems that demand homogeneity and genetic stability across cell lines.

From a biotechnological perspective, the uniformity of DSE-derived lines represents a major advantage for *Agrobacterium*-mediated transformation and subsequent recombinant protein expression, as the absence of a callus phase reduces somaclonal variability and supports stable transgene integration. This makes DSE a promising strategy for developing homogeneous embryogenic cell lines and tissues that serve as reliable biofactories for molecular farming applications.

### 2.2. Indirect Somatic Embryogenesis

Indirect somatic embryogenesis (ISE) refers to the formation of somatic embryos from callus tissue [25] (Figure 2 and Figure 3). Callus, or callus tissue, is defined as a mass of undifferentiated, rapidly proliferating cells that arise as a morphogenetic response under in vitro conditions [17]. This type of SE originates from multicellular structures, which introduces potential risks regarding genetic fidelity in regenerated plants [17]. The term “indirect” denotes the intermediate callus formation from explants before somatic embryo development [13]. In the early stages of plant tissue culture (PTC), callus formation was considered undesirable due to its association with somaclonal variation [26,27]. This term describes genetic and epigenetic alterations occurring in cells or tissues under in vitro conditions [26]. However, recent advances have repurposed somaclonal variation for genetic improvement and the selection of plants with tolerance or resistance to biotic and abiotic stressors [28,29,30].

The ISE typically involves callus induction using high concentrations of auxins, such as 2,4-dichlorophenoxyacetic acid (2,4-D), which may lead to abnormalities in the resulting somatic embryos [31]. Nevertheless, such anomalies are infrequent, as the maturation and germination phases of somatic embryos are conducted in media devoid of 2,4-D or with reduced concentrations [32]. To date, successful ISE protocols have been reported in various plant species, including garlic (*Allium sativum* L.) [33], beach naupaka (*Scaevola sericea* Vahl) [34], olive (*Olea europaea* L.) [35], Korean pine (*Pinus koraiensis Sieb*. et Zucc.) [36], and oil plant (*Camellia oleifera* Abel.) [23], among others.

On the other hand, callus formation from specific explants has also been exploited for the establishment of cell suspension cultures, leveraging the friability of the callus [37]. Friability refers to the ease with which callus cells can be dissociated (separation of the integrated cells). In agitated liquid media enriched with nutrients and growth-promoting substances, these dissociated cells can be cultured to form somatic embryos [17,38]. Additionally, this technique facilitates the production of secondary metabolites [39,40]. In recent years, cell suspension systems have gained importance in genetic engineering, providing a versatile platform for the transformation and regeneration of genetically modified cells [41,42]. Callus formation followed by somatic embryogenesis has been widely employed in plant genetic transformation aimed at recombinant protein production [43]. Through these processes, plants can serve as biofactories for the expression and accumulation of heterologous proteins, paving the way for advanced applications in biotechnology and biopharmaceuticals. The following section explores the strategies and methodologies involved in the production, recovery, and purification of recombinant proteins expressed in plants [44,45].

These results collectively suggest that the use of high auxin concentrations for callus induction must be followed by a carefully staged withdrawal to ensure normal embryo development. Although callus-mediated regeneration introduces a risk of somaclonal variation [26,27], this variation can be exploited as a source of epigenetic and genetic diversity for stress tolerance or improved secondary metabolism [28,29,30]. In *Camellia oleifera*, ISE was successfully used to obtain embryogenic callus from cotyledonary explants, leading to stable plantlet regeneration [23]. The friability of the embryogenic callus enabled the establishment of liquid cell suspension cultures, which are valuable for scaling up both clonal propagation and recombinant protein production.

From a comparative standpoint, ISE remains the most versatile and scalable route for genetic transformation because the callus phase facilitates direct access to competent cells for *Agrobacterium*-mediated transformation, biolistic delivery, or nanoparticle-based gene transfer. Friable embryogenic callus allows efficient penetration of vectors and uniform transgene integration across a large population of totipotent cells [37,41,42]. This property explains why most plant molecular farming systems, such as those developed in *Nicotiana tabacum*, *Oryza sativa*, and *Daucus carota* are derived from ISE-based cell suspensions [43]. Moreover, ISE-derived suspension cultures can serve dual functions: (i) continuous regeneration of genetically modified plantlets, and (ii) large-scale biosynthesis of recombinant proteins and secondary metabolites in controlled bioreactors [39,40]. Such systems enable cost-effective and reproducible production of vaccines, antibodies, and therapeutic enzymes, which aligns with the global shift toward sustainable, plant-based biofactories for high-value biopharmaceuticals. In summary, indirect somatic embryogenesis, despite its inherent risk of genetic instability, offers unmatched flexibility and adaptability for biotechnological applications. Through optimization of hormonal regimes, staged auxin withdrawal, and integration with suspension culture technology, ISE has evolved from a basic regeneration method into a cornerstone of plant-based molecular farming.

## 3. Plant Genetic Engineering Tools and Heterologous Protein Production

Plant-based systems have established themselves as efficient platforms for heterologous protein production. This is mainly attributed to their cellular machinery, which enables post-translational modifications, their low cultivation costs, and their capacity to accumulate proteins in specific organs, thereby facilitating product recovery and preservation [46]. Consequently, they represent a competitive alternative to systems based on bacteria, yeast, and even mammalian cells. In this context, SE constitutes an essential tool for regenerating transformed tissues and establishing molecular farming platforms [47]. Furthermore, advances in plant genetic transformation methods have accelerated the development of vaccines and antibodies with direct applications in animal and human health (Figure 4).

### 3.1. Designing Transgenic DNA Constructs

Recombinant protein production through SE, as in other plant systems, begins with the identification and amplification of the gene of interest, which is obtained from the pathogen’s DNA or RNA through PCR or RT-PCR, respectively [46]. For vaccine development, the amplified targets are usually those that encode viral capsid and surface proteins, virulence factors, microbial toxins, and other highly immunogenic peptides [48]. Advances in synthetic biology have made it unnecessary to obtain the pathogen or its nucleic acids directly. With the genomic information available in public databases, specific coding sequences can be identified in silico and subsequently synthesized de novo, base by base. The sequence must be optimized at the codon level and inserted into an initial cloning vector (plasmid). Sequence optimization at the codon level consists of adjusting the codons of the gene of interest to match the codon preference of the host species in which the protein will be expressed [49]. The DNA sequence is then verified by sequencing before being transferred to an expression vector using restriction enzymes and ligases.

Vectors for plant expression have multiple regulatory elements that determine not only transcription efficiency, but also tissue specificity and protein accumulation at certain stages of development. Among the most widely used elements are constitutive promoters, such as those from Cauliflower Mosaic Virus35S (CaMV35S) and its derivatives, as well as ubiquitin promoters (e.g., Ubi1 and Ubi2 from corn, UbiU4 from *Nicotiana sylvestris*) and actin promoters (e.g., Act1 from rice and Act2 from *Arabidopsis thaliana*), which provide constant expression in most tissues [50,51]. Organ-specific promoters are also used, such as TA29 from tobacco anthers, hsp17 from barley activated in petioles and stem xylem, Bp10 from Brassica specific to pollen, To-bRB7 from tobacco roots, and glutenin promoters (*Glu-1 and Glu-3)* with affinity for endosperm [52,53]. This diversity of promoters allows proteins to accumulate in storage organs and facilitates their recovery and applications. In addition, the introduction of modular cloning systems, such as Golden Gate cloning, has enabled the efficient assembly of multigene cassettes, facilitating the simultaneous expression of several proteins or the reconstruction of complete metabolic pathways in plants [54,55,56].

Once the coding sequence has been optimized and cloned, the next critical stage involves the rational design of the expression construct, since the architecture of the vector directly determines the transcriptional strength, stability, and accumulation of recombinant proteins in SE-derived systems. The efficiency of recombinant protein production in somatic embryogenesis (SE)-based systems depends largely on the precision of the genetic construct design. Although the workflow from gene identification to vector assembly has become standardized, recent developments in synthetic biology and codon optimization algorithms have significantly enhanced expression efficiency across plant hosts. Comparative analyses show that codon optimization can increase expression levels by up to tenfold, particularly when using species-specific preferences for *Oryza sativa* and *Nicotiana tabacum* [49]. This approach not only improves translation kinetics but also stabilizes mRNA secondary structures and reduces unintended splice sites, contributing to higher recombinant yields. The choice of promoter is equally critical, as it determines both spatial and temporal expression patterns. Constitutive promoters such as CaMV35S and ubiquitin (Ubi1/Ubi2) are preferred for preliminary proof-of-concept experiments because they drive high expression in most tissues [50,51]. However, their ubiquitous activity can lead to metabolic burden, epigenetic silencing, or interference with endogenous pathways during long-term culture. In contrast, organ- or development-specific promoters (e.g., TobRB7 for roots, Glu-1 for endosperm, and hsp17 for stress-induced expression) provide tighter control and allow protein accumulation in storage tissues, thereby simplifying purification and increasing stability [52,53]. Comparatively, the selection between constitutive and organ-specific promoters must balance yield and physiological compatibility a key factor when targeting SE-derived systems that transition through embryogenic, callogenic, and vegetative stages. For example, storage organ promoters have been effectively used in rice and maize embryos to localize recombinant proteins within protein bodies, improving downstream recovery and minimizing proteolysis. Another notable advance is the use of modular and multigene assembly systems, such as Golden Gate and Gateway cloning, which enable the construction of complex expression cassettes containing multiple regulatory units [54,55,56]. These technologies facilitate metabolic pathway reconstruction, co-expression of molecular chaperones, and glycoengineering modules that improve protein folding and post-translational modification. Such modular systems are particularly relevant for SE-based expression, as they allow parallel integration of selectable markers and morphogenic regulators (e.g., WUSCHEL and BABY BOOM) into a single construct, enhancing embryogenic competence and transformation efficiency. In summary, the design of plant expression vectors has evolved from empirical cloning strategies to highly rational, systems-based approaches. The integration of bioinformatics, synthetic biology, and modular assembly tools now allows tailored optimization of gene constructs according to the developmental and cellular context of SE-derived platforms, ensuring higher reproducibility, yield, and biosafety in the production of recombinant biopharmaceuticals.

### 3.2. Methods of Plant Genetic Transformation

Nuclear transgenesis has been the traditional and most versatile approach to plant transformation, as foreign DNA integrates into the nuclear genome. This strategy allows gene expression to be finely regulated through different promoters and enables the targeting of recombinant proteins to specific subcellular compartments. However, this system can exhibit variability in expression levels due to epigenetic silencing and Mendelian segregation [57]. Plastid transformation, in turn, offers unique advantages, such as extraordinarily high accumulation levels, maternal inheritance that minimizes gene flow to wild populations, and the possibility of introducing multiple genes as operons, similar to prokaryotic systems [58]. The most widely used method for plant transformation is orchestrated by *Agrobacterium tumefaciens*, a bacterium capable of transferring DNA fragments to plants through its T-DNA system. T-DNA is introduced into host cells through a type IV secretion system (T4SS) and integrates into the plant genome [59]. Natively, this system consists of a DNA fragment found in the Ti plasmid (tumor inducer), which contains genes for the synthesis of auxins, cytokinins, and opins, causing tumor (galls) formation [60]. In genetic engineering, the T-DNA genes are replaced by transgenic constructs. To achieve transformation, the expression vector is first introduced into *Agrobacterium* using techniques such as electroporation or conjugation, generating transformed bacterial strains that are co-cultured with plant explants or immature somatic embryos [61].

In plant species less susceptible to *Agrobacterium*, particle bombardment or biolistics is the most common alternative. Biolistics involves the direct delivery of DNA or RNA into plant cells using tungsten or gold microprojectiles coated with genetic material [62]. The microprojectiles are accelerated to high speed by a gas pressure device or a controlled explosion, allowing them to penetrate the cell walls and membranes [63]. Biolistic delivery is also widely used for transient expression, where the introduced DNA remains episomal and is expressed for a limited period without genomic integration [64]. In this context, it represents a versatile method for rapid and high-level transient expression assays, particularly useful for evaluating gene constructs, promoter efficiency, and recombinant protein accumulation prior to the establishment of stable transgenic lines [62].

Emerging new nanomaterials have had an impact on plant transformation. Carbon nanotubes (CNTs) can efficiently bind to nucleic acids and are used to deliver them to plant cells [65]. Gold nanoparticles (gold nanospheres, gold nanorods, gold nanoclusters; AuNPs) are biocompatible and can encapsulate DNA, protecting it from degradation and promoting gene expression without requiring genome integration [66]. Other inorganic nanoparticles used efficiently in obtaining transgenic plants are those composed of Ag, ZnO, TiO_2_, and CeO_2_ [67,68]. By contrast, lipid nanoparticles (solid lipid nanoparticles; SLNs, nanostructured lipid carriers; NLCs, lipid–drug conjugates; LDCs, and lipid nanocapsules; LNCs) are composed of a mixture of ionizable cationic lipids, auxiliary lipids (such as DSPC and cholesterol), and pegylated lipids [69]. These organic particles are particularly effective at protecting the genetic construct from enzymatic degradation, facilitating its transport through endocytosis, promoting endosomal escape, and increasing stable genome integration [70,71].

Each transformation method presents specific strengths and trade-offs that must be considered when designing SE-based expression systems. *Agrobacterium*-mediated transformation remains the gold standard for stable nuclear integration due to its relatively low copy number insertion, predictable expression, and minimal tissue damage [61]. However, its efficiency is highly genotype-dependent, and many monocot species, such as cereals, exhibit natural resistance to *Agrobacterium* infection. In such cases, biolistic delivery provides a robust alternative, enabling direct DNA transfer to embryogenic callus or immature somatic embryos with high reproducibility [62,63]. Nevertheless, the random integration patterns and potential for DNA fragmentation introduced by biolistics can cause transcriptional silencing or unstable expression, issues that must be mitigated by vector optimization and screening [62]. Plastid transformation, although technically more demanding, offers a unique opportunity for high-yield production of recombinant proteins with maternal inheritance and no gene flow through pollen. This is particularly valuable in SE-derived systems, where multiple embryos can be clonally propagated without the risk of transgene segregation [58]. On the other hand, the emergence of nanoparticle-mediated transformation represents a paradigm shift, providing non-integrative yet efficient delivery of nucleic acids while minimizing tissue stress [65,66,67,68,69,70,71]. Nanocarriers such as gold, silica, or lipid nanoparticles enable transient or semi-stable expression without reliance on bacterial vectors, making them ideal for regulatory-friendly biopharmaceutical production. These systems also facilitate co-delivery of multiple DNA or RNA molecules and are being explored for transient expression directly in somatic embryos, avoiding callus intermediates and reducing somaclonal variation [66,67,68,69,70,71]. Collectively, these approaches illustrate that the integration of traditional and emerging transformation technologies with SE can yield flexible and scalable biofactory systems. Selecting the appropriate method depends on the biological context: *Agrobacterium* and plastid transformation are preferred for stable, high-fidelity expression, whereas biolistic and nanoparticle-based strategies are suited for transient or rapid prototyping of recombinant constructs. As transformation technologies converge with SE-based regeneration, plant platforms are increasingly positioned to rival microbial and mammalian systems in producing next-generation therapeutic proteins.

### 3.3. Purification and Analysis of Recombinant Proteins

Once transgenic lines are established and recombinant protein expression has been achieved, attention shifts to the recovery and purification of the target product. This phase is often the most technically demanding and economically determining step of the entire production pipeline, as downstream processing largely dictates final yield, purity, and bioactivity.

After transgene expression, protein isolation begins with the harvesting of plant tissue, followed by cell disruption, extraction, and clarification of the crude extract [72,73]. Pretreatments are often applied to reduce cell debris and plant metabolites, followed by protein purification techniques. Among these isolation methods, chromatography is the most widely employed. Affinity chromatography offers high selectivity through interaction with specific tags or ligands from recombinant proteins. Ion exchange chromatography separates proteins according to their net charge and allows the removal of contaminants with different electrostatic properties. Size exclusion chromatography discriminates molecules based on their molecular mass and is particularly effective for separating aggregates or smaller peptides [74,75]. The confirmation of the identity and structural integrity of the protein is the next step, and for this purpose, mass spectrometry (e.g., MALDI-TOF and LC-MS/MS) and spectroscopic techniques (e.g., dichroism and nuclear magnetic resonance) are widely used [76]. Verification of post-translational modifications, such as folding, disulfide bridge formation, and glycosylation, is also important [77]. When the goal is vaccine development, it must be verified that the antigenic regions retain the immunogenicity of native pathogens through in vitro testing and animal models. These aspects have been thoroughly reviewed by Cirilo et al. [78].

Another benefit of producing recombinant proteins in plants is that the tissues themselves can be used as delivery vehicles, which significantly reduces purification costs. In this approach, leaves, seeds, tubers, or powders derived from transgenic biomass act as protective matrices that release the antigen into the gastrointestinal tract, a strategy particularly useful for plant-based edible vaccines [10,79,80].

In plant-based systems and particularly in SE-derived cultures or embryos, downstream processing presents unique challenges compared to microbial or mammalian platforms. The complex matrix of plant tissues contains high levels of secondary metabolites, polysaccharides, and proteases that can interfere with protein recovery and stability [72]. Consequently, extraction and clarification procedures must be carefully optimized according to tissue type and biochemical composition to preserve protein integrity and activity. Among available purification strategies, affinity chromatography remains the preferred method for recombinant proteins carrying affinity tags such as His_6_, FLAG, or Strep-tag II, as it enables selective recovery with minimal purification steps [74,75]. However, the scalability and cost of chromatography-based methods continue to limit industrial applications. To address this, alternative capture systems such as aqueous two-phase extraction and elastin-like polypeptide (ELP) tags have emerged as cost-effective, non-chromatographic approaches compatible with SE-derived suspensions. These phase-transition systems can simplify downstream processing and have been shown to reduce purification costs by up to 70% in pilot-scale studies [75].

The analytical validation of recombinant proteins through techniques such as mass spectrometry, NMR, and circular dichroism provides essential confirmation of folding fidelity and post-translational modifications [76]. This step is particularly critical for glycoproteins, since plant-specific N-glycan structures may differ from those of human origin, influencing both therapeutic efficacy and immunogenicity [77]. Beyond the molecular level, SE-derived systems offer a practical advantage through the possibility of using plant tissues themselves as delivery matrices. The concept of edible vaccines exemplifies this, where somatic embryos, seeds, or lyophilized powders act as natural encapsulation vehicles, protecting antigenic proteins from degradation and releasing them in the gastrointestinal tract without extensive purification [10,80]. This strategy substantially reduces downstream processing costs, enhances thermostability, and broadens accessibility to low-cost biopharmaceuticals.

Ultimately, the refinement of downstream processing in SE-based systems demands an equilibrium between purity, functionality, and economic feasibility. The integration of scalable purification technologies with engineered tissue types, combined with advances in glycoengineering and tissue-specific expression, positions somatic embryogenesis as a pivotal platform for the next generation of plant-made biopharmaceuticals, bridging experimental innovation with industrial applicability.

## 4. Somatic Embryogenesis for Producing Recombinant Proteins with Biomedical Applications

Several successful applications of SE have demonstrated its potential as an effective and scalable system for recombinant protein expression in plants. SE allows the regeneration of whole plants or embryogenic tissues from single somatic cells, providing a clonal and genetically stable source of material suitable for transformation and protein production. Because embryogenic cultures can be maintained over long periods in suspension, they facilitate controlled production under bioreactor conditions and enable compliance with Good Manufacturing Practices (GMP) standards [81].

For instance, recombinant proteins have been successfully produced via indirect somatic embryogenesis (ISE) in carrot (*Daucus carota* L.) expressing the hepatitis B surface antigen [82], and in maize (*Zea mays* L.) expressing the immunodominant projection domain of the infectious bursal disease virus (IBDV) for potential edible vaccine formulations [83]. The ISE system has also enabled the mass production of somatic embryos expressing the heat-labile enterotoxin B subunit (LTB) *of E. coli* in Siberian ginseng [9]. More recently, SE-derived suspension cultures of rice (*Oryza sativa*), tobacco (*Nicotiana tabacum*), and carrot have been used to produce complex therapeutic proteins, including human serum albumin (HSA), α1-antitrypsin (AAT), monoclonal antibodies (mAbs), and human growth hormone (hGH) [84,85,86,87,88,89,90,91,92,93,94,95,96,97,98]. These systems demonstrate that embryogenic cell lines can function as plant cell factories, capable of secreting correctly folded and post-translationally modified proteins with mammalian-like glycosylation. Additional studies on SE for the production of recombinant proteins with biomedical applications are summarized in Table 1.

The diversity of plant systems employing somatic embryogenesis (SE) for recombinant protein production highlights both its adaptability and scalability. As summarized in Table 1, SE-based platforms have been successfully applied across a wide range of species from herbaceous crops such as *Daucus carota* and *Medicago sativa* to woody perennials like *Juglans* and *Malus,* encompassing both direct and indirect SE pathways. These examples demonstrate how species-specific optimization of embryogenic cultures enables the production of vaccines, antibodies, and therapeutic enzymes under reproducible in vitro conditions. Among these, the most notable success is the SE-based suspension culture of carrot cells used to produce the commercially available recombinant enzyme taliglucerase alfa for the treatment of Gaucher disease [81]. This achievement exemplifies the biomedical potential of SE systems for generating high-value biopharmaceuticals, and several other plant-derived recombinant proteins have already advanced to clinical trials.

The information presented above reveals a clear and consistent trend toward the implementation of SE-derived suspension cultures as preferred platforms for the large-scale. This shift reflects not only the technical maturity of SE-based systems but also their ability to generate uniform, genetically stable cell populations amenable to bioreactor cultivation. Among the model species, *N. tabacum* and *O. sativa* stand out as dominant hosts due to their well-characterized embryogenic lines, ease of transformation, and compatibility with both transient and stable expression systems [92,93,94,95]. These features have made them indispensable for the commercial production of antibodies, enzymes, and vaccines under controlled and regulatory-compliant environments.

Conversely, the successful implementation of SE in species such as *Daucus carota*, *Eleutherococcus senticosus*, and *Medicago sativa* [9,84,92] demonstrates the adaptability of this approach beyond model plants. These species highlight the dual utility of SE systems: on one hand, their use in the production of edible vaccines and nutraceuticals, where the biomass itself serves as the delivery matrix; and on the other, their potential for synthesizing therapeutic proteins that require precise folding and post-translational modification. The diversity of molecular targets ranging from subunit antigens (CTB, LTB, HBsAg) to complex glycoproteins (hGH, HSA, mAbs) underscores the biochemical plasticity of embryogenic tissues, capable of sustaining recombinant expression without compromising cell differentiation or morphogenic capacity [84,96,97,98]. Taken together, these findings consolidate somatic embryogenesis as a unifying technological platform that bridges fundamental plant morphogenesis with industrial biomanufacturing. Its capacity to integrate genetic stability, regenerative autonomy, and biosynthetic competence distinguishes SE from conventional plant cell culture systems, positioning it as an increasingly relevant tool in molecular farming [12,13]. However, despite these achievements, the translation of SE-based systems into fully standardized production pipelines still faces significant biological and engineering challenges. Variability in embryogenic competence, limitations in transformation efficiency, and the control of protein yield under bioreactor conditions remain critical bottlenecks that must be addressed to ensure reproducibility and scalability.

## 5. Biomedical Potential of Somatic Embryogenesis Systems

Somatic embryogenesis has emerged as a promising biotechnological system for producing recombinant proteins with biomedical relevance. SE offers a morphogenetically competent tissue capable of stable genetic transformation, post-translational processing, and large-scale culture in bioreactors, combining the biosynthetic capabilities of whole plants with the uniformity and scalability of in vitro systems. These advantages position somatic embryos as suitable “green bioreactors” for the synthesis of therapeutic proteins, vaccine antigens, and other pharmacologically active macromolecules [99,100].

### 5.1. Biomedical Applications of Recombinant Proteins Derived from Somatic Embryogenesis

Early pioneering works demonstrated that somatic embryos of alfalfa (*Medicago sativa*) can accumulate higher levels of recombinant proteins than vegetative organs, achieving up to 0.15–0.18% of total soluble protein for the cholera toxin B subunit (CTB) and human interleukin-13 (hIL-13), both of biomedical importance as mucosal immunogens and cytokines [84]. Similarly, transgenic somatic embryos of Siberian ginseng (*Eleutherococcus senticosus* (Rupr. & Maxim.)) successfully expressed the *Escherichia coli* heat-labile enterotoxin B subunit (LTB), a potent mucosal adjuvant, with yields of approximately 0.36% of total soluble protein in a 130 L air-lift bioreactor, confirming the feasibility of pilot-scale production of plant-derived oral vaccines [9]. Recent studies expanded this approach to tree species. Somatic embryos of walnut (*Juglans regia* L.) were engineered to express the receptor-binding domain (RBD) and ectodomain of the SARS-CoV-2 Spike protein, achieving 3–14 µg/g dry weight while retaining structural integrity and antigenic potential [85]. This demonstrates that SE-based systems can be applied to the rapid production of diagnostic and vaccine antigens, offering an alternative to conventional microbial or mammalian cell factories, particularly for decentralized or resource-limited biomanufacturing.

### 5.2. Bioactive Metabolites and Integrated Biomanufacturing Potential

Beyond recombinant proteins, SE-derived cultures support the biosynthesis of bioactive secondary metabolites with recognized pharmacological and therapeutic activities. Pilot-scale SE cultures of *E. senticosus* yielded eleutherosides with immunomodulatory and neuroprotective properties [101], while *Catharanthus roseus* (L.) G.Don, SE lines have been exploited to enhance the production of antineoplastic alkaloids such as vincristine and vinblastine [102]. These examples reinforce the dual role of SE systems in generating both macromolecular and low-molecular-weight bioactives of biomedical significance. Recent reviews emphasize that transgenic plant systems including SE and bioreactor-grown cultures, enable efficient production of recombinant proteins and metabolites under controlled, scalable, and contamination-free conditions, offering an environmentally sustainable alternative for the bio-pharmaceutical industry [103,104]. Overall, SE-derived platforms combine the advantages of plant molecular farming with the totipotent and stable nature of embryogenic tissues, providing a versatile and low-cost route to produce vaccines, therapeutic proteins, and complex natural products within a unified plant-based biomanufacturing framework.

### 5.3. Antibodies and Immunomodulatory Proteins

Plant-derived systems obtained through somatic embryogenesis have also been employed for the production of monoclonal antibodies and immunoglobulin fragments with high therapeutic potential. For example, embryogenic cell cultures and regenerants of *Hevea brasiliensis* (Willd. ex A.Juss.) Müll.Arg. have been transformed to express foreign proteins in the lactiferous vessels, opening opportunities for antibody or vaccine antigen production directly in latex-exuding tissues [100]. These systems ensure genetic stability and continuous protein secretion, reducing purification complexity. Additionally, the controlled production of cytokines and antimicrobial peptides in somatic embryos has been proposed as an alternative for generating bioactive molecules with immunomodulatory properties [105]. Such proteins could be harvested from embryogenic tissues or used in situ for topical and oral therapeutic formulations.

### 5.4. Recombinant Proteins for Biopharmaceutical and Veterinary Use

Beyond human health, recombinant protein technology has been applied to biomedical and veterinary biotechnology, providing molecules for diagnostics, therapy, and prevention. Likewise, cloning and embryonic technologies in cattle have enabled the expression of transgenic proteins of nutritional and pharmacological relevance, paving the way for animal bioreactors producing therapeutic proteins in milk or blood plasma [106]. In microbial systems, *Escherichia coli* remains a key host for high-yield recombinant production of industrial and pharmaceutical proteins, including somatostatin, insulin, and monoclonal antibodies [107]. However, limitations in post-translational modification highlight the advantages of plant-based embryogenic systems, which can achieve proper folding and glycosylation similar to mammalian hosts but at a fraction of the cost. Recent advances have demonstrated that plant biofactories can also produce recombinant veterinary biopharmaceuticals such as subunit vaccines, antibodies, and diagnostic antigens effective against zoonotic pathogens, supporting the One Health approach that integrates animal, human, and environmental health [108]. Collectively, these advances illustrate the growing potential of somatic embryogenesis as a platform for recombinant protein production. From reproductive hormones to therapeutic enzymes and antibodies, SE-derived biofactories can sustain the synthesis of complex molecules that meet biomedical quality standards while maintaining environmental and economic sustainability.

## 6. Challenges of Somatic Embryogenesis for Recombinant Protein Production

The establishment of ES protocols as a platform for recombinant protein production faces numerous biological, technical, and regulatory challenges. Among the most influential factors are biological (e.g., genotype, cell type), physiological, and environmental (e.g., pH, temperature, light conditions) aspects that condition the embryogenic competence of tissues [109].

One of the major challenges in somatic embryogenesis-based transformation systems is their strong genotype dependence, as the embryogenic response varies considerably among species and cultivars, necessitating the design of species-specific protocols. Over the past decades, however, methodological advances have provided rational frameworks and standardized principles that facilitate the optimization of SE protocols for a given plant genotype. Another limitation arises from somaclonal variation generated during prolonged in vitro culture, which can compromise both genetic stability and the consistency of transgene expression [110,111]. Moreover, transformation efficiency remains a critical bottleneck. The stable introduction of transgenes through *Agrobacterium*-mediated or physical methods such as biolistics often shows variable success rates depending on the cell type and the embryogenic stage targeted [112]. Random integration events and positional effects within the genome further contribute to fluctuating expression levels or even gene silencing. To overcome these issues, recent developments in vector design and targeted gene-editing technologies have significantly improved transformation precision and expression stability [113,114]. For example, chloroplast transformation has emerged as a viable alternative to nuclear integration, effectively avoiding transcriptional silencing and enhancing recombinant protein yields. Likewise, the application of homologous recombination enables site-directed gene insertion, reducing the risk of disrupting vital genes and ensuring consistent, high-level expression without inducing phenotypic abnormalities.

Even when somatic embryos are successfully induced, their maturation, germination, and conversion into viable seedlings pose a challenge. Although the use of synthetic auxins, such as 2,4-D, promotes initial induction, it is associated with the formation of abnormal embryos and regeneration difficulties [31]. Maturation treatments using abscisic acid, osmotic agents (e.g., polyethylene glycol), and controlled desiccation periods have been employed to counteract these effects. These treatments promote the development of morphologically normal embryos with greater germination capacity [115,116].

Scaling up recombinant protein production at the bioreactor level introduces significant complexity due to oxygenation limitations, shear stress, and cellular heterogeneity, all of which affect productivity and reproducibility. One of the first aspects to address these challenges lies in the rational design of bioreactors capable of ensuring uniform cell growth, illumination, nutrient flow, and controlled agitation. Experimental optimization is also indispensable, requiring systematic testing of photoperiod, medium composition, and agitation parameters to guarantee consistent protein yields. To overcome issues of shear sensitivity and mass transfer, several engineering and biological strategies have been implemented [117,118]. These include the adoption of air-lift and wave bioreactors that minimize mechanical damage while improving oxygen transfer efficiency, as well as continuous or perfusion culture systems that maintain cells in exponential growth. Optimization of nutrient feeding strategies, particularly carbon and nitrogen sources, has been shown to sustain high metabolic activity and enhance recombinant protein titres [117]. In addition to physical and process improvements, the application of biochemical elicitors such as methyl jasmonate, salicylic acid, or chitosan has successfully stimulated recombinant protein accumulation by activating secondary metabolism and stress–responsive pathways [114,115,116]. Advances in metabolic and glycoengineering have further refined protein folding, stability, and secretion, notably through the inclusion of endoplasmic reticulum retention signals (e.g., SEKDEL) to prevent cytoplasmic degradation and the incorporation of specific glycosylation motifs to ensure proper functionality [98,99]. This is particularly relevant for viral glycoproteins designed as vaccine candidates, where plant-specific glycosylation can even enhance immunogenicity. However, major bottlenecks remain, including proteolytic degradation of secreted proteins, scale-dependent oxygen and nutrient gradients, and the inherent cost of bioreactors and consumables required for large-scale production. In this context, low-cost solutions such as disposable and 3D-printed bioreactors are gaining attention for their affordability and adaptability. Moreover, the containment of cells within closed bioreactor systems offers the advantage of a controlled and contamination-free environment compared with open systems. Despite persistent challenges, the ongoing integration of bioprocess modeling, advanced sensor technologies, and dynamic control of culture conditions represents a promising pathway toward achieving consistent, scalable, and economically viable production of therapeutic recombinant proteins in somatic embryogenesis-based plant systems [96,119].

From a vaccine development point of view, the most critical challenge to SE in plant systems is the low recombinant antigen yield compared to microbial, insect, and mammalian systems [120]. To overcome this issue, a high regeneration rate of a given selected plant could be associated with biomass production containing the recombinant antigen and impacting yield efficiency. In this regard, plant selection is crucial; however, genetic engineering tools available to manipulate and transform genetically must be considered, which should ideally be plant species-specific [121]. Additionally, another tactic to increase antigen yields is based on the selection and use of elements in the genetic construction, including strong promoters, codon optimization of the antigen-derived gene, and an accurate terminator, which can favor gene transcription and, consequently, antigen translation [122]. In the context of transformation techniques, *Agrobacterium*-mediated is an easy, cheap, and the most widely used biological method. However, it enables random genomic insertion of transgenes, which could be a disadvantage when it interferes with genes involved in metabolism [60]. In contrast, the biolistic method is one of the most used but expensive because it requires specialized technicians and equipment; nonetheless, the benefit is a target-specific plastid genomic transformation, avoiding undesirable transgene locations [62]. Both methods can be used in two strategies named stable and transitory, being the latter the most efficient to produce antigens, especially if the transcription is driven by an inducible promoter (instead of a constitutive promoter) [123]. Moreover, the advent of CRISPR technology has opened prospects in the field of plant-made vaccines that should be considered [124]. It should be noted that each recombinant protein produced by SE must always be evaluated on a case-by-case basis.

These challenges reflect the overall complexity of integrating SE into commercial recombinant protein production platforms. Combining advances in developmental biology, genetic engineering, cultivation and processing technologies, as well as an initial design focused on regulation and industrial scalability, is required to overcome these limitations.

## 7. Conclusions

Somatic embryogenesis (SE) is a powerful biotechnological strategy applied across numerous plant species and, in certain cases, integrated with genetic transformation protocols to produce recombinant biopharmaceutical proteins, including subunit vaccines and therapeutic enzymes. This process involves a genetically coordinated reprogramming mechanism triggered by internal and external cues, regulated through epigenetic pathways, and exploitable for the synthesis of recombinant proteins. Over the past decades, remarkable innovations have enhanced SE-based platforms for the development of plant-made pharmaceuticals. Notable examples include the production of the heat-labile enterotoxin B subunit (LTB) of *Escherichia coli* in *Eleutherococcus senticosus* somatic embryos, the hepatitis B surface antigen (HBsAg) in carrot, and monoclonal antibodies in *Nicotiana tabacum* and rice suspension cultures. Recent technological advances, such as CRISPR-mediated transformation, glycoengineering for human-compatible glycosylation, and nanomaterial-assisted DNA delivery, have expanded both the precision and efficiency of SE-based expression systems. Complementary improvements in bioreactor-scale optimization and continuous perfusion cultures have also enabled higher recombinant protein titres with reduced production costs. The commercial success of plant-made vaccines against COVID-19 and influenza, together with the *Daucus carota* suspension cell-derived enzyme taliglucerase alfa (Pfizer’s ELELYSO™, Pfizer Inc., New York, USA) for the treatment of Gaucher’s disease, provides tangible evidence that somatic embryogenesis-based systems can serve as viable platforms for affordable biotherapeutic production.

These achievements collectively position SE as a sustainable and economically accessible technology for the large-scale manufacture of vaccines, antibodies, and other therapeutic proteins, particularly relevant for developing countries. Looking ahead, the integration of SE with systems biology, omics-guided design, and artificial intelligence for process control is expected to further enhance predictability and yield. The convergence of these technologies paves the way toward next-generation plant bioreactors capable of producing safe, effective, and affordable biologics, representing a realistic pathway to democratize access to precision biopharmaceuticals for global disease prevention and treatment.

## Figures and Tables

**Figure 1 biotech-14-00093-f001:**
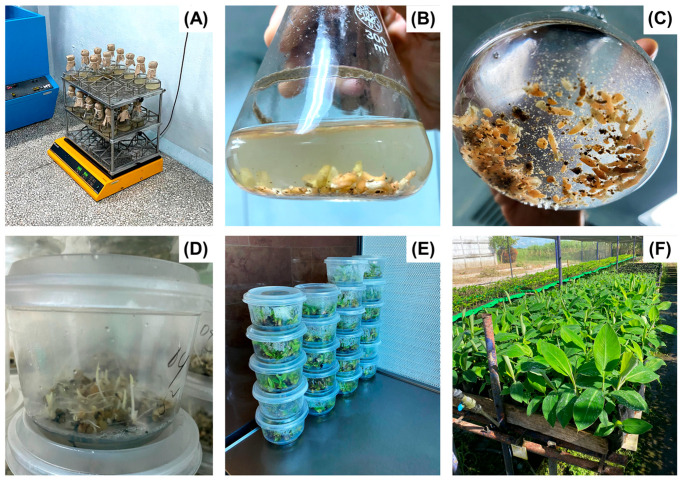
Direct somatic embryogenesis of *Musa* sp. (**A**) Formation of somatic embryos in a liquid culture medium under constant agitation; (**B**,**C**) Histodifferentiation of somatic embryos in a liquid culture medium; (**D**) Maturation of somatic embryos in a semi-solid culture medium; (**E**) Germination of somatic embryos in a semi-solid culture medium; (**F**) Conversion of germinated somatic embryos. Source: own elaboration.

**Figure 2 biotech-14-00093-f002:**
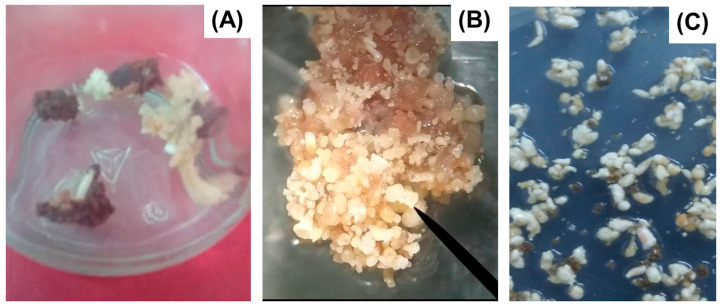
Direct somatic embryogenesis of *Coffea arabica* L. cv. ‘Caturra Rojo’. (**A**) Induction of somatic embryos; (**B**) histodifferentiation of somatic embryos in a semi-solid culture medium; (**C**) maturation of somatic embryos.

**Figure 3 biotech-14-00093-f003:**
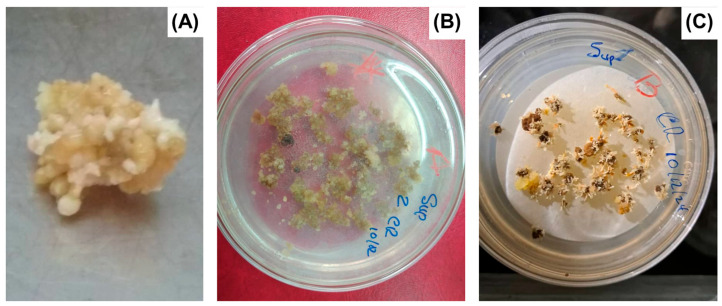
Direct somatic embryogenesis of *Coffea canephora* L. (**A**) Induction of somatic embryos, (**B**) Histodifferentiation of somatic embryos in semi-solid culture medium, (**C**) Maturation of somatic embryos. Source: own elaboration.

**Figure 4 biotech-14-00093-f004:**
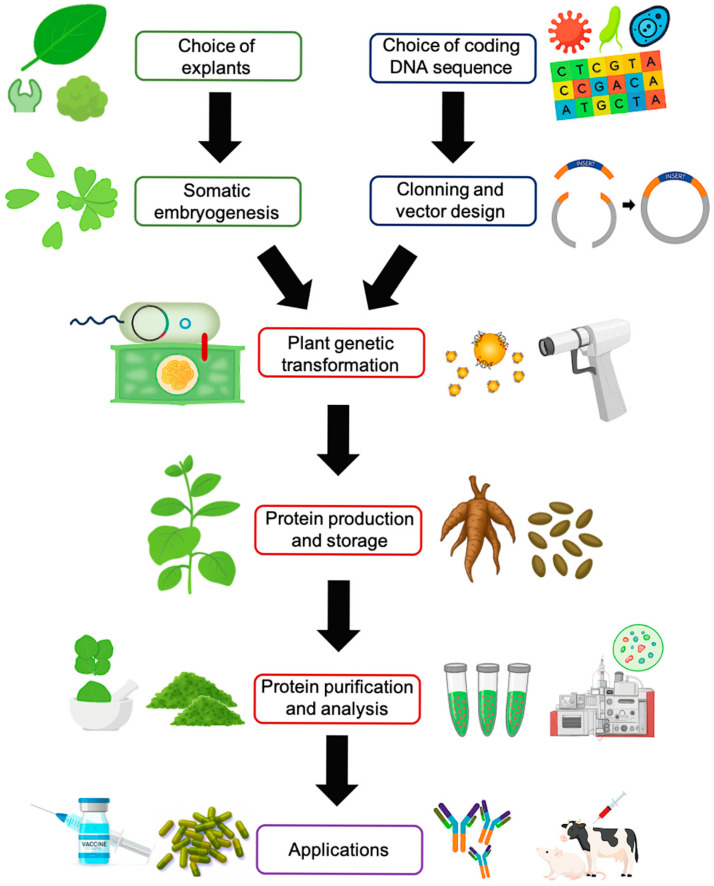
Heterologous protein production in plants through somatic embryogenesis. This process integrates molecular cloning, *Agrobacterium*-mediated or biolistic transformation, in vitro regeneration, targeted accumulation in different organs, protein recovery and purification, and, finally, the selection of the most suitable delivery vehicles for vaccines and other applications. Source: own elaboration.

**Table 1 biotech-14-00093-t001:** Somatic embryogenesis (SE) in several plant species is used for the production of recombinant proteins with biomedical applications.

Plant Species (Crop)	Protein/Product	SE System	Application(s)	Reference
*Medicago sativa* L. (alfalfa)	β-glucuronidase (GUS); cholera toxin B subunit (CTB); human interleukin-13 (hIL-13)	Indirect, from embryogenic callus derived from *Agrobacterium*-transformed plants	Oral vaccine (CTB), therapeutic (hIL-13), reporter (GUS)	[84]
*Juglans* spp. hybrid Paradox J1 (J. hindsii × J. regia)	RBD and Spike ectodomain of SARS-CoV-2; betanin	Repetitive direct (REC) without callus; embryos in DKW; *Agrobacterium* transformation	Antigens for diagnostics/vaccines; antioxidant food pigment	[85]
*Malus domestica* cv. ‘Gala’ (apple)	Reporter genes GUS and GFP; MdPDS editing (CRISPR/Cas9)	Leaves; SE mediated by auxins	SE platform for transformation/gene editing and germplasm improvement	[86]
*Daucus carota* L. (carrot)	Cholera toxin B subunit (CTB)	Indirect, from callus in hypocotyls; *Agrobacterium* transformation	Oral vaccine antigen (cholera)	[87]
*Ananas comosus* cv. Shenwan (pineapple,)	AcSERK2 (receptor-like kinase 2)	Induced with 2,4-D in basal leaf callus; unicellular origin	Early marker of embryogenic competence; role in stress response	[88]
*Gossypium hirsutum* cv. ‘Coker 315′ (cotton)	GhPLA1 (chimeric AGP, PL1 domain); AGP fractions	Indirect SE from hypocotyls: callus with 2,4-D + kinetin	SE promoter; improved regeneration and transformation	[89]
*Asparagus officinalis* L. cv. Y6 (asparagus)	Dehydrodiconiferyl alcohol (DDCA; neolignan); enzyme AoPOX1	Embryogenic callus suspension in MS ± 2,4-D	Role in cell division/differentiation (via neolignans/DCG) and as lignin precursor	[90]
*Daucus carota* L. (carrot)	Transcription factors CAREB1/CAREB2 (bZIP; binding to ABRE of Dc3 promoter)	Somatic embryos in MS in 35S:CAREB1 lines (*Agrobacterium* transformation)	ABA/sucrose-dependent regulatory framework of SE; role in maturation/dormancy	[91]
*Eleutherococcus senticosus* (Siberian ginseng)	Heat-labile enterotoxin B subunit of *E. coli* (LTB)	Transgenic somatic embryos obtained by *Agrobacterium* transformation of embryogenic cells	Antigen/adjuvant for edible vaccine; continuous production platform in bioreactor	[9]
*Nicotiana tabacum* cv. BY-2 (tobacco)	Various biotherapeutics (e.g., Taliglucerase alfa; HAS and hGH)	Suspension cultures derived from callus/embryogenic tissue	Recombinant protein production (vaccines, antibodies, and therapeutic enzymes) in GMP-compatible plant cell systems	[92]
*Oryza sativa* (rice)	Various biotherapeutics (e.g., mAbs and HBsAg)	Suspension cultures derived from callus/embryogenic tissue	Recombinant protein production (vaccines, antibodies, and therapeutic enzymes) in GMP-compatible plant cell systems	[92]
*Daucus carota* (carrot)	Various biotherapeutics (e.g., GM-CSF and hGH)	Suspension cultures derived from callus/embryogenic tissue	Recombinant protein production (vaccines, antibodies, and therapeutic enzymes) in GMP-compatible plant cell systems	[92]
*Oryza sativa* (rice)	Human serum albumin (HSA)	Transgenic suspension culture derived from callus/embryogenic tissue	Hypoalbuminemia/pharmaceutical	[93]
*Oryza sativa* (rice)	Human α1-antitrypsin (AAT)	Transgenic suspension derived from embryogenic callus	Emphysema (replacement therapy)	[94,95]
*Nicotiana tabacum* cv. BY-2 (tobacco)	Human erythropoietin (EPO)	BY-2 suspension line (derived from callus)	Tissue protection/therapeutic	[96]
*Nicotiana tabacum* cv. BY-2 (tobacco)	Human growth hormone (hGH); Human interferon α2b (IFN-α2b)	BY-2 suspension line (derived from callus)	Hormone therapy; Antiviral/immunomodulator	[97]
*Saccharum* spp. hybrids (sugarcane)	Bovine lysozyme (BvLz)	Embryogenic callus derived from rolled leaf disks and callus	Production of antimicrobial enzymes for food/cosmetic/agricultural use	[98]

## Data Availability

The original contributions presented in this study are included in the article. Further inquiries can be directed to the corresponding author.

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
