# Peer review of "Somatic Embryogenesis: A Biotechnological Route in the Production of Recombinant Proteins"

_biotech, 2025, doi:10.3390/biotech14040093_

Round 1
Reviewer 1 Report
Comments and Suggestions for Authors
The manuscript discusses the emerging importance of somatic embryogenesis as a promising approach for producing recombinant proteins in plants. The process primarily involves the formation of somatic embryos from somatic tissues and gives rise to complete plants. The process has been widely used for plant regeneration (genetically modified) and large-scale clonal propagation.
In the era of drug discovery and development, the process has shown promising results in the production of therapeutic proteins, demonstrating key advantages. The scope of the manuscript has broader scientific interests and discusses an innovative area of therapeutic production.
Some suggestions and queries for the improvement of the manuscript are discussed:
Line 27-29: This review addresses the application of SE……………. perspectives in the field.
The abstract provides a summary of the whole manuscript. It is suggested to provide key examples, challenges, and feasible solutions to large-scale recombinant protein production via SE.
Line 57-58: However, challenges remain in establishing……………recombinant DNA technology. The critical issues being discussed, what are the current innovations developed and employed to address these issues? Discuss.
Table 1. Somatic embryogenesis (SE) in several plant species for the production of recombinant 295 proteins with biomedical applications. The content of the tables is all mixed up and requires realignment for clarity.
- Challenges of Somatic Embryogenesis for Recombinant Protein Production
Line 323: Increasing production at the bioreactor levels…………… affects productivity. The large-scale production of recombinant proteins requires parameters and process optimization for minimizing losses and improving production. In this area, what are the current practices adopted to achieve a higher titre of therapeutic proteins? Are there any bottlenecks? Explain.
Conclusion section: This section seems to be a compilation of the existing information on SE-based biopharmaceutical production. No new information is being presented. It is crucial to summarize the content with the new innovative approaches in therapeutic protein production, giving specific examples. What is the future direction?
Line 363-365: Over the past decades, biotechnological advances have driven innovations…………for disease prevention. This section needs to be further elaborated to include examples of affordable medical alternatives for disease prevention and treatment.
Comments on the Quality of English Language
Moderate English revisions are required to improve clarity and readability.
Author Response
Reviewer 1
The manuscript discusses the emerging importance of somatic embryogenesis as a promising approach for producing recombinant proteins in plants. The process primarily involves the formation of somatic embryos from somatic tissues and gives rise to complete plants. The process has been widely used for plant regeneration (genetically modified) and large-scale clonal propagation.
In the era of drug discovery and development, the process has shown promising results in the production of therapeutic proteins, demonstrating key advantages. The scope of the manuscript has broader scientific interests and discusses an innovative area of therapeutic production.
Some suggestions and queries for the improvement of the manuscript are discussed:
1.- Line 27-29: This review addresses the application of SE……………. perspectives in the field.
R= We thank the reviewer for this helpful suggestion. The abstract has been revised to include representative examples of SE-based recombinant protein production systems, along with a brief mention of current challenges such as scalability, transformation efficiency, and process optimization. It now also outlines feasible strategies such as the use of bioreactor-based suspension cultures and modular vector design that address these limitations and highlight the future perspectives of SE in plant molecular farming. (lines 18-37)
2.-The abstract provides a summary of the whole manuscript. It is suggested to provide key examples, challenges, and feasible solutions to large-scale recombinant protein production via SE. Line 57-58: However, challenges remain in establishing……………recombinant DNA technology. The critical issues being discussed, what are the current innovations developed and employed to address these issues? Discuss.
R= We thank the reviewer for this constructive observation. We agree that, in addition to listing the main bottlenecks of somatic embryogenesis (SE) for recombinant protein production, it is important to briefly highlight the technological innovations that are currently being used to address them. Accordingly, we have substantially expanded the last paragraph of the Introduction to (i) explicitly link each critical issue (embryo survival, plantlet regeneration, transformation efficiency, and antigen yield) to specific mitigation strategies and (ii) provide a concise overview of the types of innovations that are later discussed in detail in Section 5 (“Challenges of Somatic Embryogenesis for Recombinant Protein Production”). In the revised version, we now mention that improvements in embryogenic competence and embryo-to-plant conversion are being achieved through optimized combinations of plant growth regulators, the use of morphogenic regulators, and maturation treatments based on abscisic acid, osmotic agents, and controlled desiccation. We also highlight recent advances in transformation efficiency, including refined Agrobacterium strains and binary vectors, physical and nanomaterial-based delivery systems, and CRISPR-based targeted integration. Furthermore, we refer to SE-derived suspension cultures operated in controlled bioreactors, glycoengineering of plant cells to obtain human-like glycosylation, and rational construct design (codon optimization, synthetic promoters and terminators, promoter stacking) as key strategies to increase recombinant protein yield and quality. These elements are now explicitly introduced in the Introduction and are further elaborated in Section 5, where each challenge is discussed together with representative technological solutions. (Lines 65-83).
3.- Table 1. Somatic embryogenesis (SE) in several plant species for the production of recombinant 295 proteins with biomedical applications. The content of the tables is all mixed up and requires realignment for clarity.
R= We appreciate the reviewer’s observation. The misalignment was caused by merged cells in the original Word file, which may have appeared disorganized during the review process. The table has now been reformatted cells have been divided and individually aligned to ensure that each row corresponds clealy to a single plant species and protein/product entry. No content has been modified; only the layout and spacing were adjusted for consistency and clarity according to the MDPI table formatting guidelines. (Table 1, lines 470-472).
4.- Challenges of Somatic Embryogenesis for Recombinant Protein Production
Line 323: Increasing production at the bioreactor levels…………… affects productivity. The large-scale production of recombinant proteins requires parameters and process optimization for minimizing losses and improving production. In this area, what are the current practices adopted to achieve a higher titre of therapeutic proteins? Are there any bottlenecks? Explain.
R= We thank the reviewer for this valuable comment. As suggested, we have expanded this section to include a detailed explanation of the current strategies employed to improve large-scale production of recombinant proteins in somatic embryogenesis (SE)-derived cultures. The new text now highlights bioprocess optimization practices such as the use of bioreactors with controlled aeration and shear stress reduction, perfusion systems for continuous culture, nutrient feed optimization, and elicitation or metabolic engineering to enhance protein yields. In addition, the main bottlenecks such as shear sensitivity, oxygen transfer limitations, and proteolytic degradation are discussed. This addition clarifies how process engineering and biological innovations are integrated to achieve higher titres of therapeutic proteins in SE-based production platforms (lines 579-597; 606-637).
5.- Conclusion section: This section seems to be a compilation of the existing information on SE-based biopharmaceutical production. No new information is being presented. It is crucial to summarize the content with the new innovative approaches in therapeutic protein production, giving specific examples. What is the future direction?
Line 363-365: Over the past decades, biotechnological advances have driven innovations…………for disease prevention. This section needs to be further elaborated to include examples of affordable medical alternatives for disease prevention and treatment.
R= We thank the reviewer for this constructive comment. The Conclusion section has been expanded to incorporate recent innovations and specific examples of therapeutic proteins and vaccines produced through somatic embryogenesis (SE) and SE-derived systems. The revised text now emphasizes new technological approaches, including CRISPR-based genome editing, glycoengineering, nanomaterial-assisted transformation, and bioreactor-scale optimization, as well as their contribution to the development of cost-effective plant-made biopharmaceuticals. The paragraph also outlines future directions and prospects for SE-based molecular farming as an affordable and scalable alternative for global health applications.(lines 665-692).

Reviewer 2 Report
Comments and Suggestions for Authors
The review titled "Somatic embryogenesis: a biotechnological route in the production of recombinant proteins" is interesting, providing a good overview of the potential of Somatic Embryogenesis (SE) and the various techniques currently employed. The authors have clearly conducted a very good bibliographic search, evidenced by the inclusion of over 110 references.
However, the manuscript currently requires substantial revision to meet the standards for a comprehensive review. While the breadth of sources is impressive, the depth of discussion is lacking. The text is currently too short relative to the number of citations provided. Many references are just briefly mentioned rather than being fully analyzed and synthesized. The authors must expand the text to critically discuss the methodologies, results, and implications of the cited works, ensuring the review adds value beyond a simple literature list.
For every figure, the authors must explicitly state the source of the image or data. If a figure is adapted or reproduced, the original publication must be clearly credited in the caption and the appropriate permission secured, where necessary.
Author Response
Reviewer 2
1.- The review titled "Somatic embryogenesis: a biotechnological route in the production of recombinant proteins" is interesting, providing a good overview of the potential of Somatic Embryogenesis (SE) and the various techniques currently employed. The authors have clearly conducted a very good bibliographic search, evidenced by the inclusion of over 110 references.
However, the manuscript currently requires substantial revision to meet the standards for a comprehensive review. While the breadth of sources is impressive, the depth of discussion is lacking. The text is currently too short relative to the number of citations provided. Many references are just briefly mentioned rather than being fully analyzed and synthesized. The authors must expand the text to critically discuss the methodologies, results, and implications of the cited works, ensuring the review adds value beyond a simple literature list.
R= We appreciate the reviewer’s thoughtful evaluation and constructive feedback. In response, we have thoroughly revised the manuscript to expand the analytical depth and improve the synthesis of cited studies throughout all major sections. The revised version now provides a more critical and integrated discussion of methodologies, comparative results, and biotechnological implications related to somatic embryogenesis (SE) and recombinant protein production.
Specifically, we have expanded the following sections to incorporate detailed comparative analyses, contextual interpretations, and critical synthesis of key literature:
Section 2.1 (lines 112–136): Enhanced with a comparative discussion of Direct Somatic Embryogenesis (DSE) across multiple species, emphasizing efficiency, genotype-dependence, and genetic stability.
Section 2.2 (lines 185–212): Expanded to include a critical comparison of Indirect Somatic Embryogenesis (ISE) protocols, hormonal regulation, and their biotechnological relevance.
Section 3.1 (lines 262–297): Strengthened with in-depth analysis of vector design strategies, codon optimization, and promoter selection, highlighting their impact on expression efficiency.
Section 3.2 (lines 341–370): Enriched with a discussion of transformation methodologies (Agrobacterium, biolistics, and nanomaterials), evaluating advantages, limitations, and applicability in SE systems.
Section 3.3 (lines 373–376; 400–432): Revised to include an analytical perspective on protein purification and validation methods, as well as challenges specific to SE-based downstream processing.
Section 4 (lines 457–468; 472–499): Expanded with a critical synthesis of SE applications in biomedical protein production, highlighting emerging trends, comparative species performance, and the scalability of SE-derived bioreactors.
Section 5 (lines 500-571) is added.
Section 5 became section 6 (lines 579-598; 606-637).
2.- For every figure, the authors must explicitly state the source of the image or data. If a figure is adapted or reproduced, the original publication must be clearly credited in the caption and the appropriate permission secured, where necessary.
R= We appreciate the reviewer’s comment. Figures 1, 2, and 3 are original and were created by the authors specifically for this review; therefore, they are of our own elaboration and do not derive from any previously published source. Figure 4 was designed and generated by the authors using Microsoft PowerPoint, based on conceptual representation and not on previously published data. All figure captions have been updated accordingly to explicitly indicate their origin. (lines 111; 156; 169; 260-261).

Reviewer 3 Report
Comments and Suggestions for Authors
Dear authors,
the article is well written, but unfortunately, it is too general. Much emphasis was placed on the process of obtaining SE. I would like the emphasis to be placed on the method of producing recombinant proteins with biomedical applications. It would also be interesting to have statistics on the number of species already used for this purpose worldwide. What else can be achieved using SE? Maybe a case study from the respective area. Develop the recombinant protein production part
Take care that the "in vitro" should be written in italics
Format references according to journal instructions.
Author Response
Reviewer 3
Dear authors,
1.- the article is well written, but unfortunately, it is too general. Much emphasis was placed on the process of obtaining SE. I would like the emphasis to be placed on the method of producing recombinant proteins with biomedical applications. It would also be interesting to have statistics on the number of species already used for this purpose worldwide. What else can be achieved using SE? Maybe a case study from the respective area. Develop the recombinant protein production part
R= We sincerely thank the reviewer for this insightful and constructive feedback. We fully agree that, beyond describing the morphogenetic basis of somatic embryogenesis (SE), the manuscript should emphasize recombinant protein production and its biomedical applications. In the revised version, we have therefore rebalanced the structure of the article and substantially expanded the sections devoted to SE-based molecular farming, as detailed below.
Stronger emphasis on recombinant protein production using SE.
We have significantly developed the section now entitled “4. Somatic embryogenesis for producing recombinant proteins with biomedical applications”. This part no longer provides only isolated examples, but explicitly discusses SE-derived systems as production platforms, highlighting their scalability, maintenance as embryogenic suspension cultures, suitability for bioreactor operation, and compatibility with Good Manufacturing Practices (GMP). We also emphasize how embryogenic lines can function as plant cell factories capable of secreting correctly folded and post-translationally modified proteins, including vaccines, antibodies, enzymes, and hormones.(lines- 457-468; 472-499).
Quantitative synthesis and “statistics” on plant species used worldwide.
To address the reviewer’s request for quantitative information, we compiled and summarized the main SE-based systems used for recombinant protein production in an updated Table 1. This table now lists more than ten plant species and genera (e.g., Daucus carota, Oryza sativa, Nicotiana tabacum, Medicago sativa, Eleutherococcus senticosus, Saccharum spp., Juglans spp., Malus domestica), indicating for each of them: (i) the recombinant protein or SE-related factor, (ii) the type of SE system (direct, indirect, or SE-derived suspension culture), and (iii) the biomedical or biotechnological application. While a comprehensive census of all SE-based platforms worldwide is beyond the scope of this review, this synthesis explicitly shows that SE has already been exploited in a broad and growing set of species for oral vaccines, therapeutic proteins, and regulatory factors.(table 1 lines 470-472).
Interpretive analysis of trends and species distribution.
The text accompanying Table 1 has been rewritten to go beyond simple listing and now includes interpretive commentary regarding species distribution and technological trends. We highlight, for example, that SE-derived suspension cultures of tobacco BY-2 and rice dominate as hosts for large-scale recombinant protein production, whereas species such as carrot, alfalfa, Siberian ginseng and walnut illustrate the adaptability of SE to non-model crops and woody perennials. This analysis clarifies how SE-based platforms are being implemented across different taxonomic groups and production contexts.
New section focusing on biomedical potential and case-based discusión (lines 500-557).
In response to the reviewer’s suggestion to explore “what else can be achieved using SE” and to include case studies, we have incorporated a new section, “5. Biomedical Potential of Somatic Embryogenesis Systems”. This section develops a case-based discussion of SE-derived systems for:
Recombinant vaccine antigens and cytokines, such as CTB and hIL-13 in alfalfa somatic embryos, the heat-labile enterotoxin B subunit (LTB) of E. coli produced in Eleutherococcus senticosus somatic embryos at pilot scale in a 130-L air-lift bioreactor, and SARS-CoV-2 Spike-derived antigens expressed in walnut somatic embryos.
Integrated biomanufacturing of bioactive metabolites, including eleutherosides in E. senticosus and antineoplastic alkaloids (e.g., vincristine and vinblastine) in Catharanthus roseus SE cultures, illustrating that SE systems can simultaneously support the production of macromolecular biotherapeutics and low-molecular-weight pharmaceutically active compounds.
Together, these examples provide the type of case-study perspective requested by the reviewer, showing how SE can be leveraged not only for recombinant proteins but also for broader biomedical and nutraceutical applications.
Global perspective on the potential of SE in molecular farming.
Finally, the end of Sections 4 and 5 now explicitly frames SE as a unifying technological platform that bridges fundamental plant morphogenesis with industrial biomanufacturing. We discuss how its integration with modern tools (synthetic promoters, CRISPR-based engineering, bioreactor technology, and glycoengineering) is progressively addressing current bottlenecks and expanding what can be achieved with SE-derived systems in the context of sustainable, plant-based biopharmaceutical production.(579-597; 606-637; 648-659).
We hope that these substantial additions and reorganizations adequately address the reviewer’s concern that the manuscript was previously too general and too focused on the SE process itself. The revised version now places a clear and explicit emphasis on recombinant protein production, biomedical applications, species coverage, and concrete case studies derived from somatic embryogenesis platforms.
2.-Take care that the "in vitro" should be written in italics
R= All instances of in vitro were corrected.
3.- Format references according to journal instructions.
R= The references were adjusted to the format requested by the journal.

Round 2
Reviewer 1 Report
Comments and Suggestions for Authors
In the current form, the manuscript may be considered for publication.